# Optimal Thermodynamic Processes For Gases

**DOI:** 10.3390/e22040448

**Published:** 2020-04-15

**Authors:** Alexei Kushner, Valentin Lychagin, Mikhail Roop

**Affiliations:** 1Faculty of Physics, Lomonosov Moscow State University, Leninskie Gory, 119991 Moscow, Russia; kushner@physics.msu.ru; 2Department of Mathematics and Informatics, Moscow Pedagogical State University, 1/1 M. Pirogovskaya Str., 119991 Moscow, Russia; 3V.A. Trapeznikov Institute of Control Sciences of Russian Academy of Sciences, 65 Profsoyuznaya Str., 117997 Moscow, Russia; valentin.lychagin@uit.no

**Keywords:** measurement, information gain, real gases, optimal control, Pontryagin’s maximum principle, Hamiltonian systems, action-angle variables, asymptotical methods

## Abstract

In this paper, we consider an optimal control problem in the equilibrium thermodynamics of gases. The thermodynamic state of the gas is given by a Legendrian submanifold in a contact thermodynamic space. Using Pontryagin’s maximum principle, we find a thermodynamic process in this submanifold such that the gas maximizes the work functional. For ideal gases, this problem is shown to be integrable in Liouville’s sense and its solution is given by means of action-angle variables. For real gases considered to be a perturbation of ideal ones, the integrals are given asymptotically.

## 1. Introduction

The problem of optimal control of thermodynamic processes has been of wide interest since the 19th century when a classical work of Carnot [1] paved the way for investigation of optimal heat engines. A number of works is devoted to constructing heat engines with maximal efficiency in case of linear heat transfer laws (see [2,3]). In [3], the problem of optimal control was investigated by means of Pontryagin’s maximum principle, which is formulated in [4] and is also described in [5]. In a relatively recent series of works [6], a non-equilibrium thermodynamic system is presented as a union of equilibrium subsystems with linear heat transfer laws between each pair of subsystems and a work of such system is maximized. Volumes of subsystems are considered to be control parameters, while state variables are entropies of subsystems. Thermodynamic optimization in some engineering problems is elaborated for example, in [7], and power optimization for irreversible thermodynamic cycles is studied in [8].

In the present work, we formulate thermodynamics as a theory of measurement of random vectors, namely extensive variables. This observation leads us to the definition of thermodynamic states as Legendrian and Lagrangian manifolds. This approach goes back to classical work [9] and is also reflected in papers [10,11]. Legendrian and Lagrangian manifolds are equipped with Riemannian structures and one of distinguishing points of this work is an observation that these structures naturally appear in measurement. This geometrical representation of thermodynamic states allows us to use Pontryagin’s maximum principle to find optimal thermodynamic process maximizing the work functional. One of the main results of this paper is that a Hamiltonian system turns out to be integrable in Liouville’s sense and we provide its exact solution. We also consider the case of real gases in virial approximation and provide commuting up to linear terms of virial expansion integrals of the Hamiltonian system for real gases.

The paper is organized as follows. In Section 2, we show relations between thermodynamics and measurement of random vectors. In Section 3, we describe Legendrian manifolds and geometric structures on them for gases in the form convenient for further optimal control problem statement. In Section 4, we state and solve the optimal control problem for ideal gases and construct asymptotics of commuting integrals for real ones.

## 2. Measurement and Thermodynamics

In this section, we briefly describe a link between thermodynamics and measurement of random vectors. Namely we show that thermodynamics can be seen as a measurement theory of extensive variables. Moreover, such a consideration leads to the notion of Legendrian manifolds representing any thermodynamic state and various geometric structures on it, in particular, Riemannian structures responsible for applicability conditions for state equations. These structures, as we shall see below, play a crucial role in control problems on Legendrian manifolds. More comprehensive discussion can be found in [12] and references therein.

### 2.1. Minimal Information Gain Principle

Let (Ω,A,p) be a discrete probability space, i.e., $\Omega =\{\omega_{1},\ldots ,\omega_{k}\}$ is a set of elementary events, A is a σ-algebra on Ω and *p* is a probability measure, $p=\{p_{1},\ldots p_{k}\}$, where $p_{i}=p\left(\omega_{i}\right)$. Let $q=\{q_{1},\ldots ,q_{k}\}$ be another probability measure equivalent to *p*. It means that measures *p* and *q* have the same zero measure sets. Introduce the *surprise function* as a random variable $s_{p}:A\to R$ by determining its values on elementary outcomes as follows:(1)$$s_{p}\left(\omega_{i}\right)=-\ln p_{i},\;i=\bar{1,k}.$$

Due to (Equation 1), we have relations $s_{p}\left(\Omega \right)=0$, $s_{p}\left(\emptyset \right)=+\infty$, therefore the notion “surprise” is justified.

The average S(p) of the surprise function $s_{p}$ with respect to the measure *p* is
(2)$$S\left(p\right)=-\sum_{i=1}^{k}p_{i}\ln p_{i}.$$

Please note that formula (Equation 2) coincides with the Shannon’s definition of entropy. If we change measure *p* to measure *q*, then we get the changing of the surprise function:$$s\left(p,q\right)=s_{q}-s_{p}=\ln\left(\frac{p_{i}}{q_{i}}\right),$$
and therefore the average of s(p,q) with respect to measure *p* called *Kullback-Leibler divergence* [13] or *information gain* is
(3)$$I\left(p,q\right)=\sum_{i=1}^{k}p_{i}\ln\left(\frac{p_{i}}{q_{i}}\right).$$

Generalization of (Equation 3) on the case of arbitrary probability space (Ω,A,p) is of the form
(4)$$I\left(p,q\right)=\int_{\Omega }\ln\left(\frac{dp}{dq}\right)dp,$$
and if dp=ρdq, where ρ is the density, then formula (Equation 4) takes the form
$$I\left(\rho \right)=\int_{\Omega }\rho \ln\rho dq.$$

Let *W* be a vector space over R, dimW=n<∞ and let X:(Ω,A,q)→W be a random vector. Let x∈W be a fixed vector, supposed to be a result of the measurement of random vector *X*, i.e., EX=x. If the initial measure *q* does not give us the required vector x∈W, then we have to choose another measure dp=ρdq, such that
(5)$$\int_{\Omega }\rho dq=1,\;\int_{\Omega }\rho Xdq=x.$$

In other words, to get a fixed vector x∈W as a result of the measurement we need to find such a density ρ that conditions (Equation 5) hold. Obviously, conditions (Equation 5) cannot determine the density ρ uniquely, therefore we need an additional requirement, which is called *the principle of minimal information gain*:(6)$$I\left(\rho \right)=\int_{\Omega }\rho \ln\rho dq\to \min_{\rho }.$$

Thus, the problem of finding the density ρ can be formulated as an extremal problem. We need to find the probability density ρ satisfying constraints (Equation 5) and minimizing functional (Equation 6).

**Theorem** **1.**
*The extremal probability measure p is given by means of density ρ as follows*
(7)$$\rho =\frac{1}{Z(\lambda )}e^{\langle \lambda ,X\rangle },\;Z\left(\lambda \right)=\int_{\Omega }e^{\langle \lambda ,X\rangle }dq,$$
*where $\lambda \in W^{*}$. The results of the measurement belong to a manifold*
$$L_{H}=\left\{x=-\frac{\partial H}{\partial \lambda }\right\}\subset W\times W^{*},$$
*where H(λ)=−lnZ(λ).*


The proof can be found in [12].

**Remark** **1.**
*The function Z*(λ)*is called the partition function*.*The function H*(λ)*is called the**Hamiltonian*.


Please note that a manifold $\Phi =W\times W^{*}$ is equipped with the symplectic structure
$$\omega =d\lambda \wedge dx=\sum_{i=1}^{n}d\lambda_{i}\wedge dx_{i}.$$

A pair (Φ,ω) is therefore the symplectic manifold. Moreover, the manifold $L_{H}$ turns out to be Lagrangian, i.e., ${\omega |}_{L_{H}}=0$.

Thus, the results of the measurement of random vectors are given by a Lagrangian manifold, and having given a Lagrangian manifold one can find out both extreme probability measure *p* and expectation *x* of random vector *X*.

Let us now introduce the information gain into the picture. To that end, construct the contactization $\hat{\Phi }$ of Φ by the following way:$$\hat{\Phi }=R\times \Phi =R^{2n+1}\left(u,x,\lambda \right).$$

Equip $\hat{\Phi }$ with the contact form
(8)$$\theta =du-\sum_{i=1}^{n}\lambda_{i}dx_{i}.$$

Thus, $\left(\hat{\Phi },\theta \right)$ is a contact space. Let $a=\left(x,\lambda \right)\in L_{H}$ and construct a manifold $\hat{L}$ of dimension *n* as follows:$$\hat{L}=\left\{u=I\left(a\right),\;x=-\frac{\partial H}{\partial \lambda }\right\}\subset \hat{\Phi }.$$

**Theorem** **2.**
*The manifold $\hat{L}$ is Legendrian, i.e., ${\theta |}_{\hat{L}}=0$.*


**Proof.** First of all, introduce a function J(x,λ):
J(λ,x)=H(λ)+〈λ,x〉.Let us show that ${J|}_{L_{H}}=I$. Indeed, using (Equation 7) we have
$${J|}_{L_{H}}=H\left(\lambda \right)\int_{\Omega }\rho dq-\langle \lambda ,H_{\lambda }\rangle =\int_{\Omega }\frac{e^{\langle \lambda ,X\rangle }}{Z(\lambda )}\left(\langle \lambda ,X\rangle -\ln Z(\lambda )\right)dq=\int_{\Omega }\rho \ln\rho dq=I.$$The differential of the function J(λ,x) is
$$dJ=\sum_{i=1}^{n}\left(x_{i}+\frac{\partial H}{\partial \lambda_{i}}\right)d\lambda_{i}+\sum_{i=1}^{n}\lambda_{i}dx_{i},$$
which implies that ${dJ|}_{L_{H}}=\hat{\theta }{|}_{L_{H}}$, where
$$\hat{\theta }=\sum_{i=1}^{n}\lambda_{i}dx_{i}.$$Taking into account the equality ${J|}_{L_{H}}=I$, we get $\hat{\theta }{|}_{L_{H}}=dI$. Finally,
$${\theta |}_{\hat{L}}={\left.\left(du-\hat{\theta }\right)\right|}_{\hat{L}}=dI-\hat{\theta }{|}_{L_{H}}=0.$$ □

It is worth saying that a canonical projection $\pi :\hat{\Phi }\to \Phi$, π(u,x,λ)=(x,λ) being restricted to the Legendrian manifold $\hat{L}$ becomes a local diffeomorphism with the image $L_{H}$, i.e., $\pi \left(\hat{L}\right)=L_{H}$ and the differential 2-form dθ is a pullback of the symplectic form ω, $d\theta =\pi^{*}\left(\omega \right)$.

Summarizing all above discussion, we conclude that any measurement of random vectors can be represented by means of Legendrian submanifold $\hat{L}$ in the contact manifold $\hat{\Phi }$. This Legendrian manifold gives us knowledge of extremal measure *p* (or, equivalently, the probability density ρ), average values *x* of random vector *X* and additionally the values of the information gain function I(λ).

### 2.2. Variance of Random Vectors

The next step is to analyze the variance of random vector *X*. Recall that the second moment is a symmetric 2-form $\mu_{2}\in S^{2}\left(W\right)$ defined by the formula
$$\mu_{2}\left(X\right)=\int_{\Omega }X\left(\omega \right)⊗X\left(\omega \right)dp.$$

*Variance* is a central second moment, i.e., a symmetric 2-form $\sigma_{2}\in S^{2}\left(W\right)$
$$\sigma_{2}\left(X\right)=\mu_{2}\left(X-\mu_{1}\left(X\right)\right)=\mu_{2}\left(X\right)-\mu_{1}\left(X\right)⊗\mu_{1}\left(X\right).$$

**Theorem** **3**[12]**.**
*The variance of a random vector X is*
$$\sigma_{2}\left(X\right)=-\mathrm{Hess}\left(H\right),$$*where $\mathrm{Hess}\left(H\right)=\sum_{i,j=1}^{n}H_{\lambda_{i}\lambda_{j}}d\lambda_{i}⊗d\lambda_{j}$ is the Hessian of the Hamiltonian H(λ).*

Please note that the symplectic manifold Φ is equipped with the universal quadratic form κ:$$\kappa =d\lambda \cdot dx=\frac{1}{2}\sum_{i=1}^{n}\left(d\lambda_{i}⊗dx_{i}+dx_{i}⊗d\lambda_{i}\right).$$

Its restriction to the Lagrangian manifold $L_{H}$
$${\kappa |}_{L_{H}}={\left.\frac{1}{2}\sum_{i=1}^{n}\left(d\lambda_{i}⊗dx_{i}+dx_{i}⊗d\lambda_{i}\right)\right|}_{x=-H_{\lambda }}=-\mathrm{Hess}\left(H\right)=\sigma_{2}\left(X\right)$$
coincides with the variance of random vector *X*. Since the variance is positive, the only areas on $L_{H}$ make sense where the differential quadratic form ${\kappa |}_{L_{H}}$ defines a Riemannian structure.

Thus, we showed that measurement of random vectors leads us to the following geometric structures on $\Phi =W\times W^{*}$.
symplectic structure
ω=dλ∧dxpseudo-Riemannian structure
κ=dλ·dx

Moreover, Lagrangian manifolds $L_{H}\subset \left(\Phi ,\omega \right)$ representing expectations of random vectors *X* consist of areas where the quadratic form ${\kappa |}_{L_{H}}$ is either positive, which we call *applicable phases*, or not.

### 2.3. Relations with Thermodynamics

First of all, we recall that any thermodynamical system is described by two types of variables, extensive (volume, energy, mass) and intensive (pressure, temperature, chemical potential). A distinctive property of extensive variables is their additivity with respect to division of the system to a disjoint union of subsystems. Secondly, the main law of thermodynamics (in particular, for gas-like systems) including the first and the second laws states that the differential form
(9)$$\theta =-dS+T^{-1}dE+pT^{-1}dV-\gamma T^{-1}dm$$
must be zero. Here *S* is entropy, *E* is energy, *V* is volume, *m* is mass, *T* and *p* are temperature and pressure respectively, γ is a chemical potential. Introducing $W_{int}=R^{3}\left(p,T,\gamma \right)$ and $W_{ext}=R^{3}\left(V,E,m\right)$ we come to a conclusion that a thermodynamical state is a Legendrian manifold $\hat{L}\subset R\times W_{int}\times W_{ext}$, where the main law of thermodynamics holds, i.e., ${\theta |}_{\hat{L}}=0$. Moreover, form (Equation 8) coincides with (Equation 9) if one puts
(10)$$du=-dS,\;\left(\lambda_{1},\lambda_{2},\lambda_{3}\right)=\left(-T^{-1},-pT^{-1},\gamma T^{-1}\right),\;\left(x_{1},x_{2},x_{3}\right)=\left(E,V,m\right).$$

Therefore, on the surface $\hat{L}$ we have the relation S=−I+α, where α is a constant. This means that thermodynamics can be viewed as a theory of measurement of extensive variables and entropy is an information gain up to a sign and additive constant. This in turn implies that principle of minimal information gain is exactly what in thermodynamics usually called *principle of maximum entropy*.

As in measurement theory, consider projection $\pi :R\times W_{int}\times W_{ext}\to W_{int}\times W_{ext}$. Then, its restriction to the manifold $\hat{L}$ gives us an immersed Lagrangian manifold $L\subset W_{int}\times W_{ext}$ and $\Phi =W_{int}\times W_{ext}$ is a symplectic space with structure form
$$\omega =d\theta =d\left(T^{-1}\right)\wedge dE+d\left(pT^{-1}\right)\wedge dV-d\left(\gamma T^{-1}\right)\wedge dm.$$

Condition for *L* to be Lagrangian is expressed as ${\omega |}_{L}=0$. Again, we can see analogies with measurement.

Pseudo-Riemannian structures coming from measurement of random vectors are inherited in thermodynamics as well. Let us define the differential quadratic form κ on $\Phi =W_{int}\times W_{ext}$ using (Equation 10) by the following way:$$\kappa =-d\left(T^{-1}\right)\cdot dE-d\left(pT^{-1}\right)\cdot dV+d\left(\gamma T^{-1}\right)\cdot dm,$$
and its restriction ${\kappa |}_{L}$ to the Lagrangian manifold *L* has to be positive. We shall see below that domains where form ${\kappa |}_{L}$ is positive correspond to phases of the medium and conditions for *L* to be Riemannian with respect to quadratic form ${\kappa |}_{L}$ are conditions of thermodynamic stability.

## 3. Legendrian Manifolds For Gases

In this section, we describe Legendrian and Lagrangian manifolds for gases (see also [14,15,16]). We pay special attention to ideal gases and virial model of real gases [17], which are used further in optimal control problem.

Let us choose the extensive variables (E,V,m) as coordinates on the Legendrian manifold $\hat{L}$. Then, on $\hat{L}$ we have entropy as a function S(E,V,m). Since entropy is an extensive quantity, the function S(E,V,m) is homogeneous of degree 1:$$S\left(E,V,m\right)=ms\left(\frac{E}{m},\frac{V}{m}\right).$$

Introducing specific variables e=E/m—specific energy, v=V/m—specific volume, s(e,v)—specific entropy, we get the following expression for contact structure θ:$$\theta =\left(-s+T^{-1}e+pT^{-1}v-\gamma T^{-1}\right)dm+\left(-ds+T^{-1}de+pT^{-1}dv\right)m,$$
on a given Legendrian manifold ${\theta |}_{\hat{L}}=0$, and therefore we get
$$-ds+T^{-1}de+pT^{-1}dv=0,\;\gamma =e-Ts+pv.$$

The differential quadratic form κ in terms of specific variables takes the form
$$\kappa =-m\left(d\left(T^{-1}\right)\cdot de+d\left(pT^{-1}\right)\cdot dv)\right),$$
and since m>0, the condition of positivity of κ becomes equivalent to negativity of the form $-m^{-1}\kappa$, which we will continue denoting by κ:(11)$$\kappa =d\left(T^{-1}\right)\cdot de+d\left(pT^{-1}\right)\cdot dv.$$

Summarizing, we have the following description of thermodynamic states of gases. Consider the contact space $\left(R^{5},\theta \right)$ equipped with coordinates (s,e,v,p,T) and structure form
$$\theta =-ds+T^{-1}de+pT^{-1}dv.$$

By a thermodynamic state we mean a Legendrian manifold $\hat{L}$, such that ${\theta |}_{\hat{L}}=0$. It can be defined by a given function σ(e,v):(12)$$\hat{L}=\left\{s=\sigma \left(e,v\right),\;p=\frac{\sigma_{v}}{\sigma_{e}},\;T=\frac{1}{\sigma_{e}}\right\}.$$

To eliminate the specific entropy form our consideration we use a projection $\pi :R^{5}\to R^{4}$, π(s,e,v,p,T)=(e,v,p,T). Its restriction to the Legendrian manifold $\hat{L}$ gives an immersed Lagrangian manifold $L\subset R^{4}$, such that ${\omega |}_{L}=0$, where
$$\omega =d\theta =d\left(T^{-1}\right)\wedge de+d\left(pT^{-1}\right)\wedge dv$$
defines a symplectic structure on $R^{4}\left(e,v,p,T\right)$. Since any 2-dimensional surface $L\subset (R^{4},\omega )$ can be given by two functions (state equations)
$$L=\left\{f_{1}\left(e,v,p,T\right)=0,\;f_{2}\left(e,v,p,T\right)=0\right\},$$
the condition ${\omega |}_{L}=0$ is expressed as $[f_{1},f_{2}]=0$ on *L*, where $\left[f_{1},f_{2}\right]$ is the Poisson bracket with respect to the symplectic structure ω:$$\left[f_{1},f_{2}\right]\;\omega \wedge \omega =df_{1}\wedge df_{2}\wedge \omega .$$

The expression for the bracket $\left[f_{1},f_{2}\right]$ in coordinates is given by the formula:$$\left[f_{1},f_{2}\right]=\frac{1}{2}\left(pT\left(f_{1p}f_{2e}-f_{1e}f_{2p}\right)+T^{2}\left(f_{1T}f_{2e}-f_{1e}f_{2T}\right)+T\left(f_{1v}f_{2p}-f_{1p}f_{2v}\right)\right).$$

Suppose that functions $f_{1}$ and $f_{2}$ are given in a usual for thermodynamics of gases form
(13)$$f_{1}=p-A\left(v,T\right),\;f_{2}=e-B\left(v,T\right).$$

Then the equation $\left[f_{1},f_{2}\right]{|}_{L}=0$ takes the form
$${\left(T^{-2}B\right)}_{v}={\left(T^{-1}A\right)}_{T}$$
and therefore the following theorem is valid

**Theorem** **4.**
*The Lagrangian manifold L is given by the Massieu-Planck potential ϕ(v,T):*
(14)$$p=RT\phi_{v},\;e=RT^{2}\phi_{T},$$
*where R is the universal gas constant.*


Using the Massieu-Planck potential one can write the differential quadratic form (Equation 11) in the following way:$$R^{-1}\kappa =-\left(\phi_{TT}+2T^{-1}\phi_{T}\right)dT\cdot dT+\phi_{vv}dv\cdot dv$$
and we conclude that conditions of applicability for the thermodynamic state model are
(15)$$\phi_{TT}+2T^{-1}\phi_{T}>0,\;\phi_{vv}<0.$$

Using (Equation 14) we obtain that inequalities (Equation 15) are equivalent to
$$e_{T}>0,\;p_{v}<0,$$
which are the conditions of thermodynamic stability.

By a *thermodynamic process* we shall mean a contact transformation of $\hat{\Phi }=R\times W_{int}\times W_{ext}=R^{5}\left(s,p,T,v,e\right)$ preserving the Legendrian manifold $\hat{L}$. Infinitesimally, such a transformation is given by a contact vector field *X*, i.e., $L_{X}\left(\theta \right)\wedge \theta =0$, where $L_{X}$ is a Lie derivative along the vector field *X*. Contact vector fields are defined by *generating functions* (see, for example, [18]) and in thermodynamic case have the form [12]:$$X_{f}=T\left(pf_{p}+Tf_{T}\right)\partial_{e}-Tf_{p}\partial_{v}+\left(f+Tf_{T}\right)\partial_{s}+T\left(f_{v}-pf_{e}\right)\partial_{p}-T\left(f_{s}+Tf_{e}\right)\partial_{T},$$
where $f\in C^{\infty }\left(\hat{\Phi }\right)$ is a generating function of the vector field $X_{f}$. One can show that $L_{X_{f}}\left(f\right)=X_{f}\left(f\right)=ff_{s}$ and therefore the vector field $X_{f}$ is tangent to the surface {f=0}. Thus, for a given Legendrian manifold $\hat{L}=\left\{f_{1}=f_{2}=f_{3}=0\right\}$ the restriction of the process $X_{f}$ to $\hat{L}$ is represented as [12]
$$X_{f}=a_{1}X_{f_{1}}+a_{2}X_{f_{2}}+a_{3}X_{f_{3}},$$
where $a_{j}$ are functions on $\hat{L}$. Using (Equation 12) we get that restrictions $Y_{j}$ of vector fields $X_{f_{j}}$ to $\hat{L}$ are
(16)$$Y_{1}=\sigma_{v}\sigma_{e}^{-2}\partial_{e}-\sigma_{e}^{-1}\partial_{v},\;Y_{2}=\sigma_{e}^{-2}\partial_{e},\;Y_{3}=0.$$

**Example** **1**(Ideal gases)**.**
*For ideal gases, the Legendrian manifold $\hat{L}$ is given by state equations*
$$f_{1}=pv-RT,\;f_{2}=e-\frac{n}{2}RT,\;f_{3}=s-R\ln\left(e^{n/2}v\right),$$*where n is a degree of freedom.*
*The differential quadratic form κ on $\hat{L}$ is*
(17)$$\kappa =-\frac{nR}{2e^{2}}de\cdot de-\frac{R}{v^{2}}dv\cdot dv.$$

*It is negative and applicable domain is therefore entire manifold $\hat{L}$.*

*Vector fields $Y_{1}$ and $Y_{2}$ have the following form*
(18)$$Y_{1}=-\frac{2ev}{nR}\partial_{v},\;Y_{2}=-\frac{2e^{2}}{nR}\partial_{e}.$$


**Example** **2**(van der Waals gases and virial model)**.**
*One of the most important models of real gases is the van der Waals model:*
$$f_{1}=\left(p+\frac{a}{v^{2}}\right)\left(v-b\right)-RT,\;f_{2}=e-\frac{n}{2}RT+\frac{a}{v}\;f_{3}=s-R\ln\left(T^{n/2}\left(v-b\right)\right),$$*where a and b are constants responsible for particles’ interaction and their volume respectively.*
*The differential quadratic form κ in coordinates (T,v) for van der Waals gases is [16]*
$$\kappa =-\frac{Rn}{2T^{2}}dT\cdot dT-\frac{v^{3}RT-2a{\left(v-b\right)}^{2}}{v^{3}T{\left(v-b\right)}^{2}}dv\cdot dv.$$

*This form can change its sign and applicable domain in a plane (T,v) for van der Waals model is given by inequality*
$$T>\frac{2a{\left(v-b\right)}^{2}}{Rv^{3}}.$$

*The virial model for real gases’ state equations was proposed in [17] and is of the form*
$$p=\frac{RT}{v}\left(1+\sum_{i=1}A_{i}\left(T\right)v^{-i}\right).$$

*For van der Waals gases, we will mainly be interested in the first term of the expansion which has the form*
$$A_{1}\left(T\right)=b-\frac{a}{RT}.$$

*In this approximation, vector fields $Y_{1}$ and $Y_{2}$ are*
(19)$$Y_{1}=-\frac{2a(ev+a)}{Rv^{2}n}\partial_{e}-\frac{2(ev+a)}{Rn}\partial_{v},\;Y_{2}=-\frac{2{\left(ev+a\right)}^{2}}{nRv^{2}}\partial_{e}.$$


## 4. Optimal Control

In this section, we formulate the control problem for thermodynamic processes of gases and provide exact solution for ideal gases and asymptotic expansion of integrals for real ones.

Let thermodynamic state of a gas be given by a Legendrian manifold $\hat{L}$ and let us choose vector fields $Y_{1}$ and $Y_{2}$ defined by formula (Equation 16) as a basis in module of vector fields on $\hat{L}$. We will use the notation x=(e,v). Let $x^{\left(1\right)}=\left(e_{1},v_{1}\right)$ and $x^{\left(2\right)}=\left(e_{2},v_{2}\right)$ be two fixed points in applicable domains on $\hat{L}$. Let $l\subset \hat{L}$ be an integral curve of the unknown vector field $Y=u_{1}Y_{1}+u_{2}Y_{2}$ and let α=pdv be a work 1-form. Introduce a quality functional *J*:(20)$$J=\int_{l}\alpha .$$

Physical meaning of *J* is a work of the gas along the process curve *l*. We are looking for a process $Y=u_{1}Y_{1}+u_{2}Y_{2}$ such that functional (Equation 20) reaches its maximum value. Vector $u=(u_{1},u_{2})$ is a vector of control parameters. If *t* is a parameter on *l*, then we will suppose that t=0 corresponds to the point $x^{\left(1\right)}$ and $t=t_{0}$, where $t_{0}$ is a given value of the parameter *t*, corresponds to $x^{\left(2\right)}$. Rewrite the vector field *Y* as
$$Y=Y^{\left(1\right)}\left(x,u\right)\partial_{e}+Y^{\left(2\right)}\left(x,u\right)\partial_{v},$$
where coefficients $Y^{\left(1\right)}$, $Y^{\left(2\right)}$ are defined by means of (Equation 16).

We define the domain of admissible control parameters by means of the differential quadratic form κ. On the Legendrian manifold its physical meaning is (up to a sign) the variance of extensive variables (e,v), we limit a relative variance by a positive number δ:$$-\frac{\kappa (Y,Y)}{e^{2}}\leq \delta ,$$
which leads to inequality
$$-\kappa \left(Y_{1},Y_{1}\right)u_{1}^{2}-2\kappa \left(Y_{1},Y_{2}\right)u_{1}u_{2}-\kappa \left(Y_{2},Y_{2}\right)u_{2}^{2}\leq \delta e^{2}.$$

Therefore, for a given point $x\in \hat{L}$, the boundary ∂U of the admissible domain *U* for control parameters is an ellipse with a centre at that point and whose semi-axes depend, in general, on *x*.

Summarizing, we formulate an extremal problem for finding the process *Y* in the form:
$$\begin{array}{cc} \dot{x}=\left(Y^{\left(1\right)}\left(x,u\right),Y^{\left(2\right)}\left(x,u\right)\right), & \;x\in R^{2},\;u\in U,\\ x\left(0\right)=x^{\left(1\right)},\;x\left(t_{0}\right)=x^{\left(2\right)}, & \\ J=\int_{0}^{t_{0}}\alpha \left(Y\right)dt\to \max_{u\in U}. & \end{array}$$

The Hamiltonian of problem (Equation 21) has the form
(21)$$H\left(x,\lambda ,u\right)=\alpha \left(Y\right)+\lambda_{1}Y^{\left(1\right)}\left(x,u\right)+\lambda_{2}Y^{\left(2\right)}\left(x,u\right),$$
where $\lambda =(\lambda_{1},\lambda_{2})$ are Lagrangian multipliers.

### 4.1. Ideal Gases

For ideal gases, vector fields $Y_{1}$ and $Y_{2}$ have form (Equation 18) and vector field *Y* is
$$Y=-\frac{2ev}{nR}u_{1}\partial_{v}-\frac{2e^{2}}{nR}u_{2}\partial_{e}.$$

Therefore using expression (Equation 17) for the differential quadratic form κ in case of ideal gases we get the domain *U* of admissible control parameters:$$U=\left\{\left(u_{1},u_{2}\right)\in R^{2}|\frac{4}{n^{2}R}u_{1}^{2}+\frac{2}{nR}u_{2}^{2}\leq \delta \right\},$$
and its boundary is an ellipse with constant semi-axes.

The commutator of vector fields $Y_{1}$ and $Y_{2}$ is
$$\left[Y_{1},Y_{2}\right]=\frac{2e}{nR}Y_{1}.$$

The dual basis is generated by 1-forms
$$\xi_{1}=-\frac{nR}{2ev}dv,\;\xi_{2}=-\frac{nR}{2e^{2}}de.$$

Due to the Lie-Bianchi theorem (see, for example, [18]), 1-form $\xi_{2}$ is exact, i.e., $\xi_{2}=dq_{1}$, where $q_{1}=nR{\left(2e\right)}^{-1}$. The restriction of the form $\xi_{1}$ to the curve $q_{1}=C_{1}$ is exact too and its potential is $q_{2}=-C_{1}\ln v+C_{2}$, where $C_{i}$ are constants. Let $q=(q_{1},q_{2})$ be new coordinates on $\hat{L}$. Then, the inverse transformation is
(22)$$e=\frac{nR}{2q_{1}},\;v=\exp\left(-\frac{q_{2}}{q_{1}}\right).$$

In new coordinates $\left(q_{1},q_{2}\right)$ vector fields $Y_{1}$ and $Y_{2}$ take the form:$$Y_{1}=\partial_{q_{2}},\;Y_{2}=\partial_{q_{1}}+\frac{q_{2}}{q_{1}}\partial_{q_{2}}.$$

Therefore Hamiltonian (Equation 21) will take the form
(23)$$H\left(q,\lambda ,u\right)=-\frac{Ru_{1}}{q_{1}^{2}}+\lambda_{1}u_{2}+\lambda_{2}\left(\frac{q_{2}u_{2}}{q_{1}}+u_{1}\right).$$

Since Hamiltonian (Equation 23) is linear with respect to control parameters $\left(u_{1},u_{2}\right)$, it reaches its extremal values on the boundary ∂U. Let τ be a parameter on ∂U. Then control parameters $\left(u_{1},u_{2}\right)$ can be written as
$$u_{1}=\frac{n\sqrt{R\delta }}{2}\cos\tau ,\;u_{2}=\sqrt{\frac{nR\delta }{2}}\sin\tau ,$$
and the Hamiltonian H(q,λ,u) takes the form
(24)$$H\left(q,\lambda ,\tau \right)=\frac{\sqrt{2nR\delta }q_{1}\left(q_{1}\lambda_{1}+q_{2}\lambda_{2}\right)\sin\tau +\sqrt{R\delta }n\left(q_{1}^{2}\lambda_{2}-R\right)\cos\tau }{2q_{1}^{2}}.$$

To find the points where the Hamiltonian H(q,λ,τ) reaches its maximum one has to resolve the equation $H_{\tau }=0$ with respect to τ:$$\sin\left(\tau +\arctan\left(\frac{\sqrt{2}q_{1}\left(q_{1}\lambda_{1}+q_{2}\lambda_{2}\right)}{\sqrt{n}\left(R-q_{1}^{2}\lambda_{2}\right)}\right)\right)=0.$$

Its solution is
(25)$$\tau^{*}\left(q,\lambda \right)=\pi \left(2k+1\right)-\arctan\left(\frac{\sqrt{2}q_{1}\left(q_{1}\lambda_{1}+q_{2}\lambda_{2}\right)}{\sqrt{n}\left(R-q_{1}^{2}\lambda_{2}\right)}\right),\;k\in Z.$$

Substituting roots (Equation 25) into (Equation 24) we get the following expression for Hamiltonian H(q,λ):(26)$$H\left(q,\lambda \right)=\frac{1}{2q_{1}^{2}}\sqrt{nR\delta \left(nq_{1}^{4}\lambda_{2}^{2}+2q_{1}^{4}\lambda_{1}^{2}+4q_{1}^{3}q_{2}\lambda_{1}\lambda_{2}+2q_{1}^{2}q_{2}^{2}\lambda_{2}^{2}-2Rnq_{1}^{2}\lambda_{2}+R^{2}n\right)}.$$

To find the optimal process, one needs to solve the system
(27)$${\dot{q}}_{1,2}=\frac{\partial H}{\partial \lambda_{1,2}},\;{\dot{\lambda }}_{1,2}=-\frac{\partial H}{\partial q_{1,2}},$$
where the Hamiltonian H(q,λ) is given by (Equation 26). Since the Hamiltonian H(q,λ) does not depend on the parameter *t* explicitly, it is the integral of system (Equation 27). Moreover, the following theorem is valid:

**Theorem** **5.**
*Hamiltonian system (Equation 27) has an integral $G\left(q,\lambda \right)=q_{1}\lambda_{2}$ which is in involution with the Hamiltonian H(q,λ) with respect to the Poisson bracket on phase space, i.e., [G,H]=0, where*
[G,H]Ω∧Ω=dG∧dH∧Ω,Ω=dq∧dλ.


Thus, Hamiltonian system (Equation 27) has two commuting integrals and is therefore integrable in Liouville’s sense.

To construct solution to (Equation 27) we use the method of action-angle variables (see, for example, [19]). The invariant manifold *M* of system (Equation 27) is given by levels $H_{1}$ and $H_{2}$ of its integrals:$$M=\left\{\left(q,\lambda \right)\in R^{4}|H\left(q,\lambda \right)=H_{1},\;G\left(q,\lambda \right)=H_{2}\right\}.$$

Choose $\left(q_{1},q_{2}\right)$ as local coordinates on *M*. Then we have
$$\lambda_{1}=\frac{-2H_{2}R\delta nq_{2}\pm \sqrt{D}}{2Rn\delta q_{1}^{2}},\;\lambda_{2}=\frac{H_{2}}{q_{1}},$$
where $D=2R\delta n\left(4H_{1}^{2}q_{1}^{4}-\delta Rn^{2}{\left(R-H_{2}q_{1}\right)}^{2}\right)$. Therefore the manifold *M* can have different numbers of connected components depending on the number of roots of polynomial *D*.

**Theorem** **6.**
*The manifold M has three connected components if levels of integrals $H_{1}$ and $H_{2}$ are related as*
$$H_{2}^{4}\delta n^{2}-64RH_{1}^{2}\geq 0.$$

*Otherwise, the manifold M has two connected components.*


Singularities of projection of *M* to the plane $\left(q_{1},q_{2}\right)$ are given as $\Sigma =\cup \Sigma_{j}$, where
$$\Sigma_{j}=\left\{\left(q_{1}^{\left(j\right)},q_{2}\right)|q_{2}\in R,\;D\left(q_{1}^{\left(j\right)}\right)=0\right\}.$$

Thus, for a given initial point $\left(q^{\left(0\right)},\lambda^{\left(0\right)}\right)$ the reachability set consists of points of *M* belonging to the same connected component as $\left(q^{\left(0\right)},\lambda^{\left(0\right)}\right)$ does.

Let us choose two Hamiltonian vector fields $X_{1}=X_{H}$ and $X_{2}=X_{G}$ as a basis in module of vector fields on phase space $R^{4}\left(q,\lambda \right)$. Here
$$X_{f}=f_{\lambda_{1}}\partial_{q_{1}}+f_{\lambda_{2}}\partial_{q_{2}}-f_{q_{1}}\partial_{\lambda_{1}}-f_{q_{2}}\partial_{\lambda_{2}}.$$

We need to find two closed 1-forms $\varkappa_{1}$ and $\varkappa_{2}$ dual to restrictions $Z_{1}$ and $Z_{2}$ of vector fields $X_{1}$ and $X_{2}$ on *M*, i.e., $\varkappa_{i}\left(Z_{j}\right)=\delta_{ij}$, where $\delta_{ij}$ is the Kronecker symbol. On each connected component of *M* the forms $\varkappa_{1}$ and $\varkappa_{2}$ are exact, i.e., $\varkappa_{i}=d\Omega_{i}$ and functions $\Omega_{i}$ are called *angles*. Expressions for $\Omega_{1}$ and $\Omega_{2}$ are given by the following theorem, which is the result of straightforward computations.

**Theorem** **7.**
*Angle variables $\Omega_{1}$ and $\Omega_{2}$ are of the form*
(28)$$\Omega_{1}=\pm \int \frac{4H_{1}q_{1}^{2}dq_{1}}{\sqrt{D}},\;\Omega_{2}=\frac{q_{2}}{q_{1}}\pm \int \frac{n^{2}R\delta \left(R-H_{2}q_{1}\right)dq_{1}}{q_{1}\sqrt{D}}.$$

*Hamiltonian system (Equation 27) is equivalent to*
$${\dot{\Omega }}_{1}=1,\;{\dot{\Omega }}_{2}=0.$$


Thus, the solution of (Equation 27) is given as
$$\Omega_{1}=t+\alpha_{1},\;\Omega_{2}=\alpha_{2},$$
where constants $\alpha_{1}$ and $\alpha_{2}$ are derived from conditions at the ends. By means of inverse transformation (Equation 22) one can obtain the corresponding solutions in terms of thermodynamic variables (e,v).

### 4.2. Real Gases

Here, we again will look for a process $Y=u_{1}Y_{1}+u_{2}Y_{2}$, where vector fields $Y_{1}$ and $Y_{2}$ are given by (Equation 19). Following the case of ideal gases, we finally get the Hamiltonian $H_{vdW}\left(q,\lambda \right)$ in the form
(29)$$H_{vdW}\left(q,\lambda \right)=H\left(q,\lambda \right)+aH_{a}\left(q,\lambda \right)+bH_{b}\left(q,\lambda \right)+\ldots ,$$
where the first order corrections $H_{a}$ and $H_{b}$ are
$$H_{a}\left(q,\lambda \right)=\frac{e^{q_{2}/q_{1}}\left(q_{1}^{2}\left(R\delta n^{3}\lambda_{2}^{2}-8H^{2}\left(q,\lambda \right)\right)-R^{2}\lambda_{2}n^{3}\delta \right)}{4q_{1}nRH\left(q,\lambda \right)},\;H_{b}\left(q,\lambda \right)=\frac{e^{q_{2}/q_{1}}R\delta n^{2}\lambda_{2}\left(R-\lambda_{2}q_{1}^{2}\right)}{4H\left(q,\lambda \right)q_{1}^{2}}.$$

We will restrict ourselves to the linear with respect to parameters *a* and *b* corrections only.

From now and on, we will assume that all the functions are expressed in terms of angle variables $\left(\Omega_{1},\Omega_{2}\right)$ given by (Equation 28) instead of $\left(q_{1},q_{2}\right)$. This can be done by resolving (Equation 28) with respect to $\left(q_{1},q_{2}\right)$. In these new coordinates, vector fields $Z_{1}$ and $Z_{2}$ have the form
(30)$$Z_{1}=\frac{\partial }{\partial \Omega_{1}},\;Z_{2}=\frac{\partial }{\partial \Omega_{2}}.$$

To integrate the Hamiltonian system with Hamiltonian (Equation 29), one needs to find the second commuting integral $G_{vdW}\left(q,\lambda \right)$. We will look for that integral in the form
$$G_{vdW}\left(\Omega_{1},\Omega_{2}\right)=G\left(\Omega_{1},\Omega_{2}\right)+aG_{a}\left(\Omega_{1},\Omega_{2}\right)+bG_{b}\left(\Omega_{1},\Omega_{2}\right)+\ldots ,$$
where functions $G_{a}$ and $G_{b}$ are to be defined. Condition $[H_{vdW},G_{vdW}]=0$ leads us (up to linear terms) to the following equations:(31)$$\left[H_{a},G\right]=\left[G_{a},H\right],\;\left[H_{b},G\right]=\left[G_{b},H\right].$$

Using a well-known relation $\left[f,g\right]=X_{g}\left(f\right)$ and (Equation 30), we get system (Equation 31) as
$$\frac{\partial H_{a}}{\partial \Omega_{2}}=\frac{\partial G_{a}}{\partial \Omega_{1}},\;\frac{\partial H_{b}}{\partial \Omega_{2}}=\frac{\partial G_{b}}{\partial \Omega_{1}}.$$
and finally we obtain
$$G_{a}=\int \frac{\partial H_{a}}{\partial \Omega_{2}}d\Omega_{1},\;G_{b}=\int \frac{\partial H_{b}}{\partial \Omega_{2}}d\Omega_{1}.$$

Thus, we have the second integral for the extremal problem commuting with the Hamiltonian up to linear in *a* and *b* terms and therefore the Hamiltonian system is integrable in Liouville’s sense in this approximation.

## 5. Conclusions

We showed that considering thermodynamics as a theory of measurement of random vectors one can describe thermodynamic states as Legendrian or Lagrangian manifolds equipped with the differential quadratic form responsible for the variance of extensive variables. Thermodynamic processes are interpreted as curves on Legendrian manifolds. The Hamiltonian system arising from the problem of finding an optimal curve maximizing the work functional is shown to be integrable in Liouville’s sense and its solution is constructed explicitly by means of action-angle variables. We also provided a method of finding asymptotically commuting integrals for real gases in virial approximation.

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
