# Peer review of "Optimal Thermodynamic Processes For Gases"

_entropy, 2020, doi:10.3390/e22040448_

Round 1

Reviewer 1 Report

Equilibrium thermodynamics of gases is formulated in terms of a measurement theory of random vectors of extensive variables using methods of differential geometry. In this approach thermodynamic states are presented as Legendrian and Lagrangian manifolds that allows one to find the optimal thermodynamic process using the Pontryagin maximum principle for the work functional. The paper contains a number of important and interesting results. For example, it is shown that the Hamiltonian arising in the geometric formulation of the principle of minimal information gain is Liouville integrable that leads to the exact solution of the corresponding Hamiltonian system.

In my opinion, the paper can be accepted for publication in Entropy in present form.

Author Response

Dear Reviewer 1,

Thank you for your attention to the paper.

Reviewer 2 Report

Nice written paper (too mathematical but readable). The subject is interesting, worth publication and broad discussion.

In introduction more broad description of Pontryagin's maximum principle will be valuable (not only reference [4]). Concluding remarks (and summary of obtained results) is needed.

The manuscript can (after small revision) can be recommended for publication.

Author Response

Response to Reviewer 2 Comments
Dear Reviewer 2,
Thank you for your attention to the paper.

Point 1: In introduction more broad description of Pontryagin's maximum
principle will be valuable (not only reference [4]).
Response 1: Pontryagin's maximum principle has become a well-known and
widely used method in optimization problems, therefore we decided not to give its
comprehensive description. We have added reference [5] (see line 14) which is
lecture notes on optimization problems, where this method is analyzed and
illustrated with some examples.

Point 2: Concluding remarks (and summary of obtained results) is needed.
Response 2: Section “Conclusions” has been added.

Reviewer 3 Report

Report on MS entropy-750442
Optimal Thermodynamic Processes For Gases
by Alexei Kushner, Valentin Lychagin, and Mikhail Roop

Dear Editor:
Authors reformulate classical equilibrium thermodynamics of gasses using notions from differential geometry and information theory. In these terms they formulate and discussan optimization problem of finding equilibrium processes that yield an optimal work value. The optimization procedure can be carried out explicitly for ideal gases. Realgases, understood here in terms of a virial expansion, are studied asymptotically.
After a brief introduction (Section 1), authors define basic geometric notions and briefly relate the constructions to equilibrium thermodynamics (Section 2). In Section 3, equations of states are used to defined manifolds for gases. The last section (Section 4)comprises formulation of the optimization procedure. The manuscript lacks conclusions (there is no concluding section) and contains no figures. The bibliography lists 15 items, 5 of which are self-citations, and the rest are prevalently classical works (e.g.by Carnot, Gibbs, Pontryagin).
Conceptually, the manuscript is not suitable for a physics journal, which Entropy in my opinion still is. Instead, the manuscript content, presentation style, and results belong to pure mathematics. Therefore I recommend to reject the manuscript and transfer it to a more suitable mathematical journal.
If, despite the above recommendation, the authors would decide to re-submit their work into a physics journal. The manuscript would require a major revision. In its current form it would hardly be appreciated by an average physicist working on optimization of thermodynamic processes. From a physicist’s perspective, the main problems with the manuscript are as follows:

• It is completely disconnected from a vivid research on thermodynamic optimization currently going on in physics and engineering literature; viz the reference list not containing any contemporary work except authors’ ones.
• Optimization of equilibrium processes for ideal gases is a classical problem that can be solved without an additional geometric formalism. Authors do not explain what are the benefits resulting from the usage of differential geometry. Without such explanations Sections 1–3 may seem rather redundant.
• As mentioned above, authors would need to focus more on practical, demanding and interesting applications. Now the major part of the manuscript is formed by a mere repetition of definitions. Are there any practical problems that can be conveniently solved using their approach, which are at the same time hardly manageable by a classical variational calculus?
• Terminology used in the manuscript is not consistent with good practices in
physics literature. What authors call “information gain” is the Kullback-Leibler
divergence, the symbol “I(p, q)” is usually reserved for the mutual information, what authors call “Hamiltonian” is nothing but the free energy. Also several important references to the standard literature, which would help the reader to orient better in the notions, are missing.
• There is no discussion of the results, their significance, and perspectives for their future impact. Even though this may be a standard way to prepare manuscripts in a pure mathematics, it is highly non-standard in physics.
To conclude, the above points does not mean that the manuscript cannot be highly appreciated by a community working in the research field where it belongs to (differential geometry). However, the manuscript is not suitable for publication in a physics journal like Entropy

Author Response

Response to Reviewer 3 Comments
Dear Reviewer 3,
Thank you for your attention to the paper. This work, as you fairly mentioned, is
essentially mathematical and therefore we decided to submit it to the special issue“Thermodynamics, Geometry and Control Theory” of Entropy.

Point 1: It is completely disconnected from a vivid research on thermodynamic
optimization currently going on in physics and engineering literature; viz the
reference list not containing any contemporary work except authors’ ones.
Response 1: Some references regarding current investigations of optimization in
thermodynamics have been added [7], [8] (lines 18-20).

Point 2: Optimization of equilibrium processes for ideal gases is a classical
problem that can be solved without an additional geometric formalism. Authors donot explain what are the benefits resulting from the usage of differential geometry. Without such explanations Sections 1–3 may seem rather redundant.
Response 2: Geometrical formalism is needed since the formulation of the
optimization problem uses Riemannian structure to define the domain of
admissible control parameters, not to mention that the state space of the
optimization problem is a Legendrian manifold.

Point 3: As mentioned above, authors would need to focus more on practical,
demanding and interesting applications. Now the major part of the manuscript is
formed by a mere repetition of definitions. Are there any practical problems that
can be conveniently solved using their approach, which are at the same time hardly manageable by a classical variational calculus?
Response 3: The work is theoretical and the main result of the paper (integrability of the Hamiltonian system) seems important itself from the theoretical point of view.

Point 4: Terminology used in the manuscript is not consistent with good practices in physics literature. What authors call “information gain” is the Kullback-Leibler divergence, the symbol “I(p, q)” is usually reserved for the mutual information, what authors call “Hamiltonian” is nothing but the free energy. Also several important references to the standard literature, which would help the reader to orient better in the notions, are missing.
Response 4: We have added the notion of Kullback-Leibler divergence and
corresponding reference (lines 55-56).

Point 5: There is no discussion of the results, their significance, and perspectives
for their future impact. Even though this may be a standard way to prepare
manuscripts in a pure mathematics, it is highly non-standard in physics.
Response 5: We have added the section “Conclusions”.

Round 2

Reviewer 3 Report

2nd Report on MS entropy-750442
Optimal Thermodynamic Processes For Gases
by Alexei Kushner, Valentin Lychagin, and Mikhail Roop

Dear Editors:
In response to my comments, authors have performed following minor changes in the
manuscript:
• Two new references were added into the introduction (Refs. [7] and [8]).
• The notion of Kullback-Leibler divergence and the corresponding reference were
added.
• The new section “Conclusion” was added.
As a reply to my requests to exemplify practical implications of authors’ formalism and explain significance of the results, the authors claim that:
The work is theoretical and the main result of the paper (integrability of the Hamiltonian system) seems important itself from the theoretical point of view.
The response letter also contains the following remark, which I assume is a reaction to my general criticism that the manuscript is not suitable for publication in a physics journal:
This work, as you fairly mentioned, is essentially mathematical and therefore we decided to submit it to the special issue “Thermodynamics, Geometry and Control Theory” of Entropy.
Otherwise, the general criticism from my first report has not been addressed.
In view of the last quoted remark, which I was unaware of when writing the first report, it can happen that the topical aims and scopes of the aforementioned special issue papers and formal requirements imposed on them could be different compared to regular articles. In this case, I recommend to consult the issue with a scientific editor of Entropy, who should decide whether the manuscript is suitable for the publication in
the special issue. My review report should be understood as a review of a manuscript submitted to Entropy as a regular article